# Novel Four-Cell Lenticular Honeycomb Deployable Boom with Enhanced Stiffness

**DOI:** 10.3390/ma15010306

**Published:** 2022-01-01

**Authors:** Hui Yang, Shuoshuo Fan, Yan Wang, Chuang Shi

**Affiliations:** 1College of Electrical Engineering and Automation, Anhui University, Hefei 230601, China; f876272482@163.com; 2College of Mechanical Engineering, Yanshan University, Qinhuangdao 066004, China; wangyan_597@163.com; 3State Key Laboratory of Robotics and System, Harbin Institute of Technology, Harbin 150001, China; ty12shichuang@126.com

**Keywords:** deployable structures, four-cell lenticular honeycomb boom, coiling dynamics, optimization, composite material

## Abstract

Composite thin-walled booms can easily be folded and self-deployed by releasing stored strain energy. Thus, such booms can be used to deploy antennas, solar sails, and optical telescopes. In the present work, a new four-cell lenticular honeycomb deployable (FLHD) boom is proposed, and the relevant parameters are optimized. Coiling dynamics analysis of the FLHD boom under a pure bending load is performed using nonlinear explicit dynamics analysis, and the coiling simulation is divided into three consecutive steps, namely, the flattening step, the holding step, and the hub coiling step. An optimal design method for the coiling of the FLHD boom is developed based on a back propagation neural network (BPNN). A full factorial design of the experimental method is applied to create 36 sample points, and surrogate models of the coiling peak moment (*M_peak_*) and maximum principal stress (*S_max_*) are established using the BPNN. Fatigue cracks caused by stress concentration are avoided by setting *S_max_* to a specific constraint and the wrapping *M_peak_* and mass of the FLHD boom as objectives. Non-dominated sorting genetic algorithm-II is used for optimization via ISIGHT software.

## 1. Introduction

Deployable composite ultra-thin booms can easily be folded and self-deployed by releasing stored strain energy; thus, these booms can be applied to membrane antennas and solar sails. Many cross-sectional deployable booms, such as lenticular booms [1], triangular rollable and collapsible (TRAC) booms [2,3], and storable tubular extendable member (STEM) booms [4,5], are available.

For example, a novel collapsible boom has been applied to a roll-out solar array, and rolling and deploying experiments have been performed on this prototype [6]. Coiling of the tape-spring and composite thin-walled booms [7,8] and the micromechanical behavior of two-ply weave laminates under small strains have also been investigated [9]. An active–passive composited driving deployable lenticular boom for space probes has been proposed. In addition, an optimal design of lenticular booms has been developed for its modal and wrapping analysis [10,11,12]. Experiments and numerical studies of the flattening and wrapping processes of deployable composite thin-walled lenticular tubes have been conducted [13,14,15,16,17]. A TRAC boom has been designed using a data-driven computational framework without considering the influence of the bonding web between two tape-springs [18,19]. The shapes of the consistent features of folded orthotropic collapsible booms made from metal and woven laminated composites have been calculated [20]. TRAC booms feature 10-fold greater cross-sectional inertia compared with lenticular booms and 34-fold greater cross-sectional inertia compared with STEM booms at the same package height [21]. An N-shaped cross-section boom has been proposed, and its post-buckling properties have been optimized [22].

A four-cell lenticular honeycomb deployable (FLHD) boom composed of four pairwise symmetrical tape-springs has been proposed, as shown in Figure 1. Here, the two outer tape-springs feature only two bonded webs and four arcs, while the two inner tape-springs have four bonded webs and eight outer tangent arcs. The FLHD boom illustrated in Figure 1c is derived from the modified double lenticular (MDL) boom shown in Figure 1b. Compared with the DL boom in Figure 1a, the middle segment of the MDL boom is a smooth section located at the junction of the two arc segments. This segment is also key to the expansion of the MDL boom into the FLHD boom.

The FLHD boom has higher bending and torsional stiffness than the DL and TRAC booms. However, material damage may occur during the dynamic process of complete coiling around the hub. Thus, geometric parameters critical to the deployment mechanism of the FLHD boom should be optimized.

When the FLHD boom is close to the holding state after the full-flattening step, its cross-section could abruptly snap. The deformation is located in a short transverse curved region, and the moment increases quickly and reaches a peak moment (Mpeak). Mpeak and mass are set as objectives, while the maximum principal stress (Smax) is set as a constraint to increase the deployment-state stiffness. The bonded web-1 (w), arc radius (r), and central angle (θ) of the tape-spring are set as variables. The finite element (FE) models of the full-coiling process are solved using the ABAQUS/Explicit solver. Non-dominated sorting genetic algorithm-II (NSGA-II) is used to obtain an optimal design.

The contents of this manuscript are as follows. The behavior of the FLHD boom is presented in Section 2.1, the three analysis steps of the complete coiling process are presented in Section 2.2, and the numerical results are discussed in Section 2.3. Surrogate models are established using a back propagation neural network (BPNN) in Section 3 and Section 4. The multi-objective optimization design is presented in Section 5. Concluding remarks are provided in Section 6.

## 2. Problem Description

### 2.1. Behavior of FLHD Booms

The geometric dimensions of the cross-section of an FLHD boom are shown in Figure 2. The FLHD boom is similar to the lenticular boom: it can be flattened and coiled and then expanded to its initial state by releasing stored strain energy. The FLHD boom is composed of four sets of two symmetrical tape-springs and has three independent parameters, i.e., the central angle *θ*, the arc radius *r*, and the bonded web-1 width *w*. The mass of the FLHD boom is written as:(1)mass(h,θ,r)=8ρ⋅L⋅t(3rθ+w)
where *t* is the thickness of the thin-walled boom.

The FLHD boom coiling mechanism is simplified to 15 parts, as shown in Figure 3. Two radial rollers are symmetrically positioned on both sides of the FLHD boom, and 11 circumferential rollers are evenly distributed around a hub. The radius *R* of the hub is 125 mm, and the radial and circumferential rollers guide the FLHD boom to coil smoothly around the hub. The axial distance between reference point (RP) 13 and RP14 is 78 mm, and the radial distance between the outer arc of the circumferential roller and the edge of the hub is 25 mm. The longitudinal length (*L*) and *t* of the FLHD boom are set to 2000 mm and 0.18 mm, respectively; the stacking sequence of the composite material is [45°/−45°/−45/°45°], as shown in Figure 4, and the thickness of each ply is *t_p_* = 0.045 mm. The ply material is T800; it is a strategic new material with low density, good rigidity, and high composite strength, which plays an irreplaceable role in aerospace and other fields. The material properties of T800 are listed in Table 1.

### 2.2. Analysis Steps

An FE model of the FLHD boom is established for numerical analysis. The FLHD boom is defined as an extruded shell consisting of 20,114 nodes and 19,214 four-node reduced integrated shell elements (S4R). The FE model is shown in Figure 5. The outer surface of the hub is connected to control point reference point 1 (RP1), and two FLHD boom nodes at the end of the bonded web are set as slave nodes. All the outer surfaces of the 11 circumferential and 2 radial rollers are connected to reference points from RP2 to RP14. The connections between the bonded web and four tape springs are modeled by the tie constraint. The interactions between the rollers and FLHD boom or between the hub and FLHD boom are constructed with surface contact as a frictionless property. The FLHD boom is divided into segments of 50 mm and 175 mm from the left end along the 3-axis. The coiling process consists of a flattening step (step time 0.1 s), a holding step (step time 0.05 s), and a hub coiling step (step time 0.5 s). No direct contact exists between the surface of the boom and the surface of the hub, and an interval of 0.36 mm is set to avoid stress concentration. The node sets of the FE model are shown in Figure 6.

In the first step of the simulation, the FLHD boom is pulled flat using tension. A shell edge load of 800N/m is applied to node-set A along the 2-axis direction, and a shell edge load in the opposite direction with the same size is applied to the B set. A viscous pressure of 0.5 Pa is applied to all surfaces of the boom to ensure the convergence of the explicit dynamic process. Two 50 N/m shell edge loads in opposite directions are applied to node sets F and G along the 3-axis to prevent the FLHD boom from warping. Node-set D and 14 reference points are clamped to fix the FLHD boom, hub, and rollers. The boundary condition of node-set D has fixed; node-sets A and B release displacement DoFs along the 1-axis and 2-axis.

In the second step, a pressure of 0.1 MPa is applied to node sets C, J, I, H, and K, and the FLHD boom end is clamped and fixed on the hub. Boundary conditions are set as same as those in the first step.

The last step is coiling one circle around the hub. The connection between the FLHD boom and hub is modeled by a beam-type multi-point constraint. RP1 of the hub is the control point, and node-set E is considered the salve node. A continuous rotation of 0.5 s is applied to RP1, the rotation displacement of UR 2 is set to 5.88 rad, and a steady step amplitude is established to reduce the loading shock on the FLHD boom. The timestep of the analysis is set to 0.1 s, and other degrees of freedom (DoFs) associated with RP1 are fixed. The DoF of node-set G is also fixed, except for the displacement along the 3-axis. Node-set G release displacement DoFs along the 3-axis. Node-set D releases displacement DoFs along the 1-axis and 3-axis, and rotate DoFs about the 2-axis.

### 2.3. Numerical Results and Discussion

The total CPU time required by an Intel(R) Core(TM) i5-9400F CPU @ 2.90GHz desktop computer to achieve one complete coil is approximately 20 h. The stress nephogram of the flattening, holding, and coiling processes for an FDLH with *θ* = 52.5°, *w* = 8 mm, and *r* = 27 mm is shown in Figure 7. In the flattening step, *S_max_* is located close to the lower smooth section of the end of the FLHD boom, as shown in Figure 7a; however, in the holding and hub coiling steps, *S_max_* is located in the middle smooth section connected to the arc sections on both sides of the boom near the hub. *S_max_* does not show a regular trend with increasing wrapping angle, and its change shows strong nonlinearity. *S_max_* during the coiling process is 632 MPa. If *S_max_* exceeds the allowable stress of the material, the FLHD boom is destroyed. Therefore, *S_max_* should be reduced. RP1 is the geometric center point of the hub, and the change of moment of RP1 during the whole coiling process can best reflect the change of moment of the whole boom. The moment curve of RP1 during the whole process is shown in Figure 8. At the beginning of the holding step, the moment increases to a peak of 30.03 Nm; it then decreases sharply and reaches a stable value of approximately 5 Nm. While the wrapping *M_peak_* can lock the deployment mechanism, if the *M_peak_* is too large, the strain energy of the FLDH boom becomes excessive. Therefore, the coiling *M_peak_* is set as the threshold.

## 3. BPNN Surrogate Model Method

### 3.1. Description of the Optimization Problem

The simulation time of the FE model is too long and finding the optimal size of the FDLH boom requires a large number of model simulations. Thus, establishing a mapping relationship between variables and targets is necessary to save time and resources. Because this procedure is complex and mathematically challenging, an artificial NN is used as a surrogate model to obtain the mapping relation.

### 3.2. BPNN Surrogate Model

The BPNN surrogate model is established to represent the mapping relationship between inputs and outputs through the NN. The response value of the point to be measured is then predicted through the mapping relationship to save time and resources. The BPNN is a multilayer feed-forward NN, and the three-layer NN structure is composed of an input layer, a hidden layer, and an output layer, as shown in Figure 9. The layer to layer is fully interconnected, and no mutual connection occurs between the same layer. The *x*_1_, *x*_2_, and *x*_3_ are the three-dimensional input vectors corresponding to the two-dimensional output vectors *y*_1_ and *y*_2_. The 3-15-2 structure is adopted [23], where *w_i,j_* represents the weight value from the *i*th node of the hidden layer to the *j*th node of the input layer, *θ_i_* represents the threshold value of the *i*th node of the hidden layer, *w_ki_* is the weight from the *k*th node of the output layer to the *i*th node of the hidden layer, *a_k_* is the threshold of the *k*th node of the output layer, and *O_k_* is the output of the *k*th node of the output layer.

The input of the *i*th node of the hidden layer:(2)$$net_{i}=\sum_{j=1}^{M}w_{i}{}_{j}x_{j}$$

The linear function can be written as:(3)$$O_{ij}=\beta (O_{ij})=\psi (O_{ij})=net_{i}+\theta_{i}$$

The sigmoid logarithm is defined as:(4)$$O_{i}=\phi (net_{i})=\frac{1}{1+e^{-net_{i}}}=\frac{1}{1+e^{-(\sum_{j=1}^{M}w_{ij}x_{j}+\theta_{i})}}$$
where *E_p_* is the error criterion function of each sample *p*, *M* is the number of output nodes, and *T_k_* and *O_k_* are the desired and actual outputs, respectively.
(5)$$E_{p}=\frac{1}{2}\sum_{k=1}^{M}{\left(T_{k}-O_{k}\right)}^{2}$$

Modifying the output layer weight and threshold correction $\Delta \omega_{ki}$ and $\Delta a_{k}$, hidden layer weight and threshold correction $\Delta \omega_{ij}$ and $\Delta \theta_{i}$ according to the gradient descent method of error, and *η* is the learning rate.
(6)$$\Delta \omega_{ki}=-\eta \frac{\partial E_{p}}{\partial \omega_{ki}},\;\Delta a_{k}=-\eta \frac{\partial E_{p}}{\partial a_{k}},\;\Delta \omega_{ij}=-\eta \frac{\partial E_{p}}{\partial \omega_{ij}},\;\Delta \theta_{i}=-\eta \frac{\partial E_{p}}{\partial \theta_{i}}$$

## 4. Build BPNN Surrogate Model of Mpeak and Smax

### 4.1. Sample Points

There are three design variables of the FLHD boom in the sample space, i.e., *θ*, *r*, and *w*, each of which varies within a specific range. The design range of all parameters is determined according to the actual situation. Considering the complexity of the running time of FE simulations, a total of 36 combinations of design sample points with different levels of the full-factor method are selected, as shown in Table 2.

### 4.2. Error Analysis of the Surrogate Model

In general, the smaller the error of the surrogate model, the greater its fit and the higher its prediction accuracy. However, in actual applications, decreases in fitting error initially decrease the prediction error but then increase the error when a certain fitting error is reached. This phenomenon is referred to as the overfitting phenomenon in the BPNN. Obtaining more data is the best way to solve this overfitting problem. If sufficient data are available, the model can identify exceptions with greater accuracy. A diagram of the fitting effect of the training sample points is shown in Figure 10. An expression of the relative error is then defined to analyze the fitting effect of the model quantitatively.
(7)$$RE=\frac{\tilde{f}(x^{\left(i\right)})-f(x^{\left(i\right)})}{f(x^{\left(i\right)})}\;\;\;\;i=1,2,\cdots m$$
where $f(x^{\left(i\right)})$ is the FE result of sample *i*, $\tilde{f}(x^{\left(i\right)})$ is the BPNN surrogate model result of sample *i*, and *m* is the number of samples.

The errors between the FE and BPNN surrogate model results of *M_peak_* and *S_max_* are shown in Figure 11. The REs of *S_max_* and *M_peak_* are approximately ±3%, which means the fitting precision meets the present requirements. Five samples are randomly generated in the sample space to test the model prediction accuracy of the BPNN surrogate model, and the REs of the tested sample points are shown in Table 3. The REs obtained do not exceed −6.3%, which meets the accuracy requirements of the current work.

### 4.3. Response Surface

The response surfaces of *M_peak_* and *S_max_* obtained when the *r* values of the FLHD boom are 23, 25, and 27 mm are shown in Figure 12. When *r* and *θ* are constant, the *M_peak_* first increases and then decreases while *S_max_* continuously increases as *w* increases from 7 mm to 9 mm. Small changes in *M_peak_* and *S_max_* are noted as *θ* increases from 52° to 60°. These results indicate that changes in *θ* have little effect, whereas changes in *w* exert remarkable effects on *M_peak_* and *S_max_*. The sensitivity of *M_peak_* and *S_max_* to *θ* is enhanced as *r* increases.

## 5. Multi-Objective Optimization Design

The coiling *M_peak_* of a boom represents its ability to resist external disturbances and lock deployable membrane antennas and solar and drag sails. When the maximum concentrated stress exceeds the allowable stress, the FLHD boom is destroyed. The mass of the modified boom directly affects the cost of launching spacecraft. Thus, *M_peak_* and mass are selected as objectives, and *S_max_* is selected as the constraint. Moreover, *θ*, *r*, and *w* are set as design variables. The multi-objective design model of the FLHD boom can be written as follows:(8)$$\left\{ \begin{array}{l} \mathrm{Opt}.\left\{M_{f}^{p}\left(w,\theta \right)\leq 35\mathrm{Nm},{\;mass|}_{\min}\right\}\\ S.t.S_{\max}\left(w,\theta \right)\leq 650\mathrm{Mpa};\\ 23\;\mathrm{mm}\leq r\leq 27\;\mathrm{mm}；\\ 7\;\mathrm{mm}\leq w\leq 9\;\mathrm{mm};\\ 52.5{}^{\circ}\leq \theta \leq 60{}^{\circ}. \end{array}\right.$$

The *Objective* value of the objective function is calculated as the weighted sum of all optimized objective components *X_i_* and the weighting factor *W_i_* and scaling factor *SF_i_* corresponding to the *i*th objective component, as shown in Equation (9).
(9)$$Objective=\sum \frac{W_{i}X_{i}}{SF_{i}}$$

If the *SF_i_* and weight of the target component are equal, larger magnitudes of the target component will have greater impacts on the optimization result. Thus, the *SF_i_* and weight are set to adjust the importance of each subtarget in the optimization process. The scaling factor *SF_i_* is uniformly set to 1, and the weight factors of *M_peak_*, *S_max_*, and *mass* are set to 1.0, 1.0, and 20, respectively, according to the order of magnitude to facilitate calculations. The Pareto fronts of *mass* and *M_peak_* in the optimization process are shown in Figure 13.

In the non-dominant sorting genetic algorithm (NSGA-II), each target parameter is processed separately. Standard genetic manipulations of mutation and hybridization perform in the design. The selection process is based on two main mechanisms: “non-dominated sorting” and “congestion distance sorting.” By the end of the optimization run, each design has a “best” combination of goals, and it is impossible to improve one goal without sacrificing one or more other goals. NSGA-II is used to carry out multi-objective optimization in Isight software. The optimization part of the paper establishes the BPNN surrogate model with Matlab, forms an input–output mapping relationship, and then uses the OPTIMIZATION module in Isight for algorithm iteration optimization. The whole loop is closed, and the whole process can be automated, as shown in Figure 14. Here, the maximum number of iterations is set to 50, the population size is set to 48, and the crossover probability is set to 0.9. The optimal result is *r* = 23 mm, *θ* = 53.31°, and *w* = 7.52 mm. The REs of *M_peak_* and *S_max_* are less than 10%, as shown in Table 4.

## 6. Conclusions

This paper describes the design, coiling dynamics analysis, and optimization of a new type of thin-walled boom. The conclusions are summarized as follows:
A novel type of FLHD boom characterized by a high spreading ratio, light weight, and simple structure is proposed.A surrogate model of *M_peak_* and *S_max_* is established by BPNN, and the REs of the *M_peak_* and *S_max_* of 36 sample points do not exceed 3%, which verifies the accuracy of the surrogate model.NSGA-II is used to complete the multi-objective optimization design. *M_peak_* and mass are selected as objectives, *S_max_* is selected as the constraint, and *θ*, *r*, and *w* are set as design variables. The optimal design structure is *r* = 23.00 mm, *θ* = 53.31°, and *w* = 7.52 mm. The REs of the optimal design results are less than −5.54%.The next step is to build an experimental platform to verify the reliability of the theory and simulation analysis. After the verification results are reliable, FLHD boom will be applied to the folding of large-aperture array antennas.


## Figures and Tables

**Figure 1 materials-15-00306-f001:**
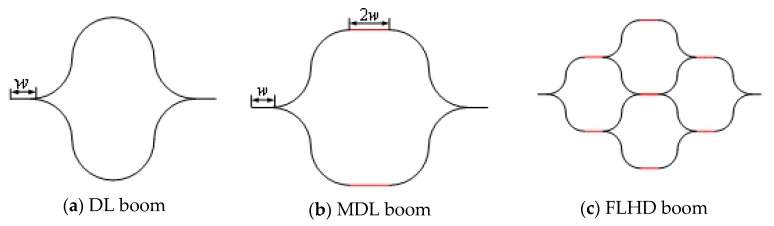
Three kinds of composite deployable booms.

**Figure 2 materials-15-00306-f002:**
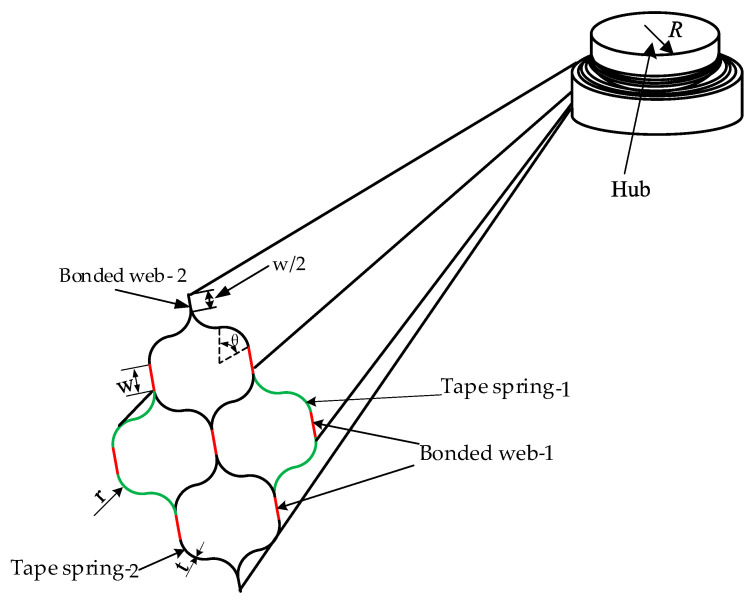
Geometric dimensions of the cross-section of the FLHD boom.

**Figure 3 materials-15-00306-f003:**
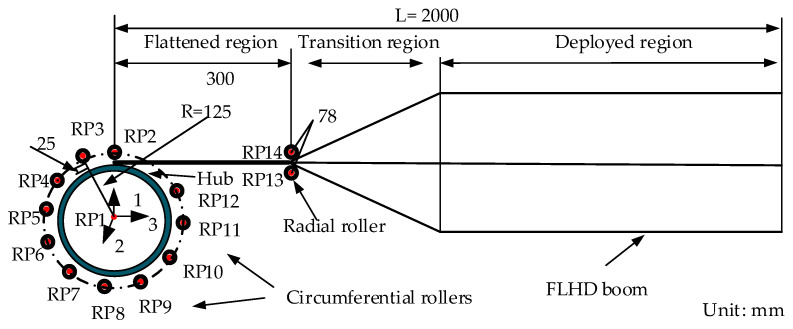
Simplified schematic of the FLHD boom coiling structure.

**Figure 4 materials-15-00306-f004:**
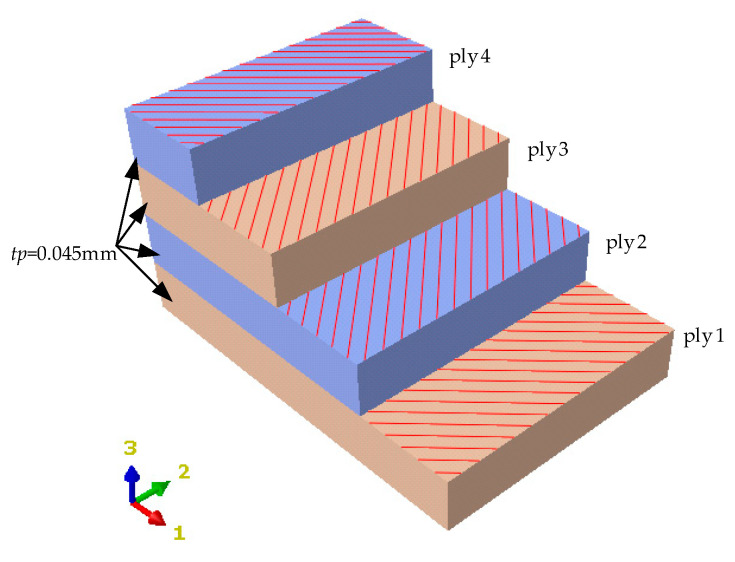
Ply angle diagram of the FLHD boom.

**Figure 5 materials-15-00306-f005:**
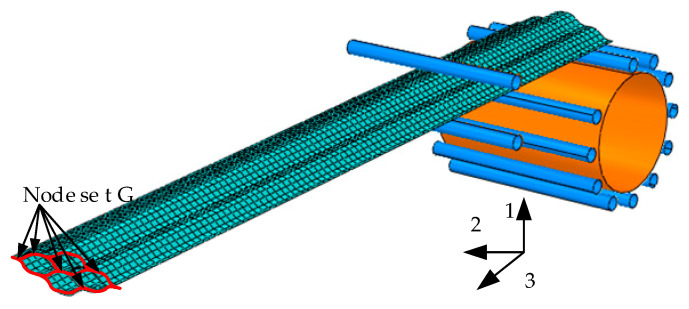
FE model.

**Figure 6 materials-15-00306-f006:**
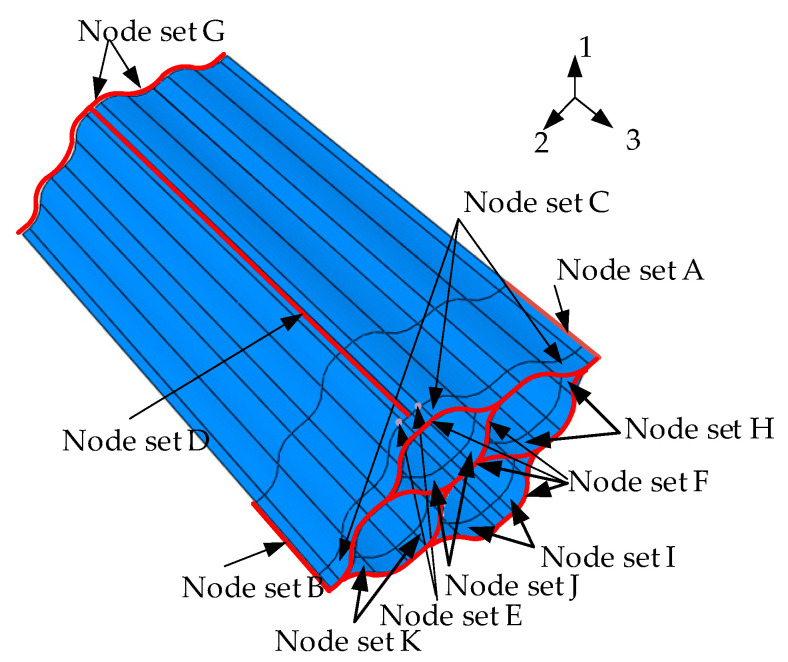
Node sets used in the FE model.

**Figure 7 materials-15-00306-f007:**
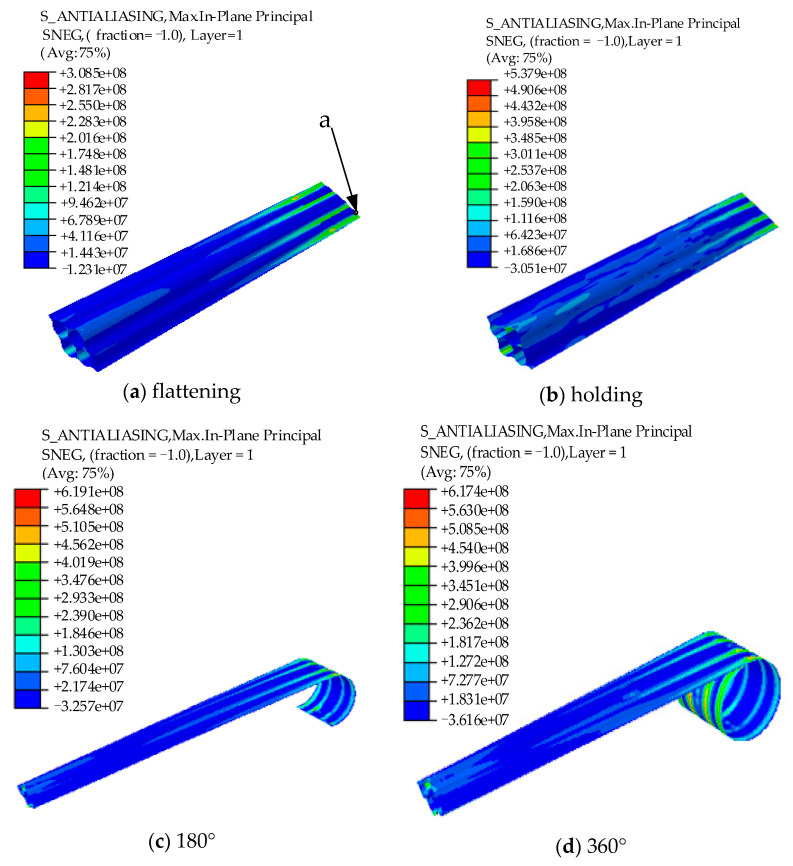
Principal stress nephogram of one FLHD boom with coiling of 360° (Unit: pa).

**Figure 8 materials-15-00306-f008:**
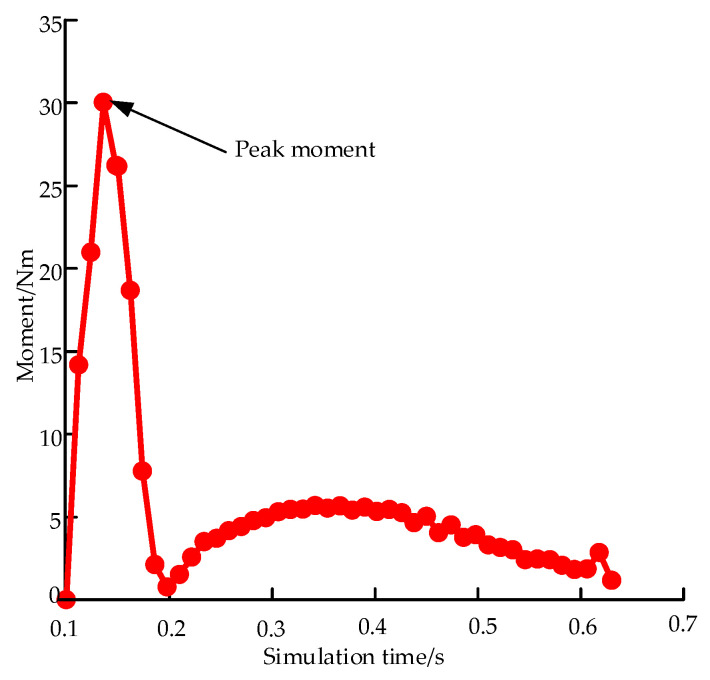
Moment of RP1 during the whole coiling process.

**Figure 9 materials-15-00306-f009:**
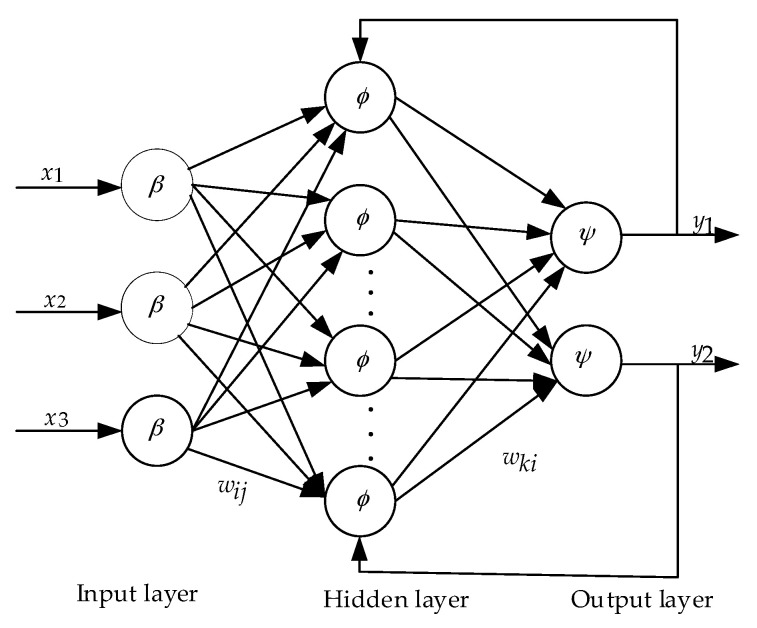
Simplified schematic of the BPNN structure.

**Figure 10 materials-15-00306-f010:**
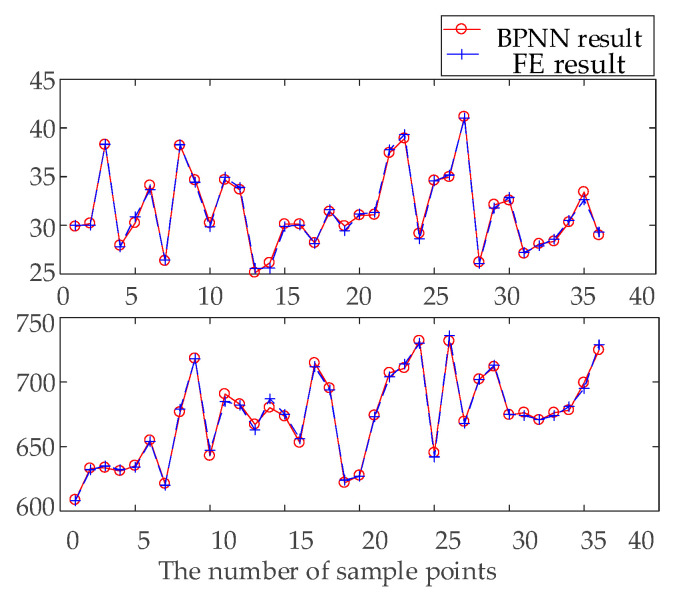
Fitting diagram of training sample points.

**Figure 11 materials-15-00306-f011:**
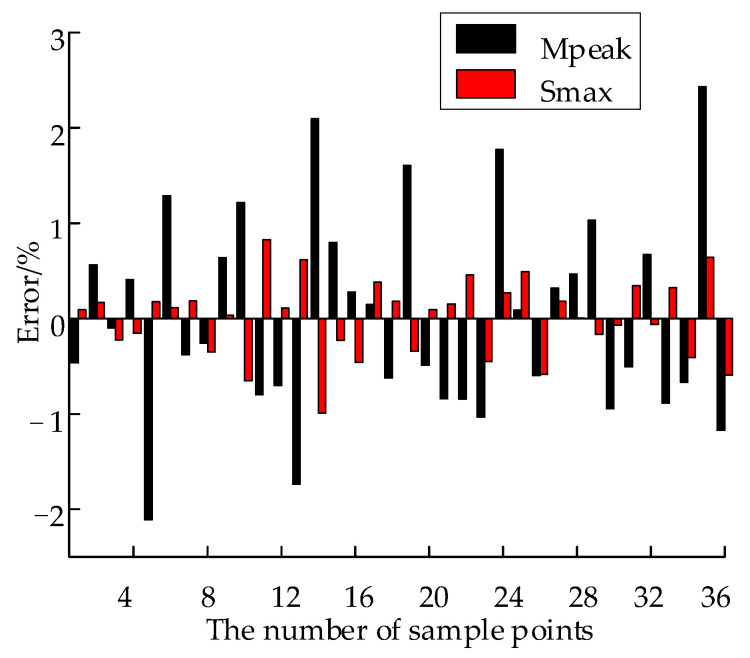
REs of *M_peak_* and *S_max_*.

**Figure 12 materials-15-00306-f012:**
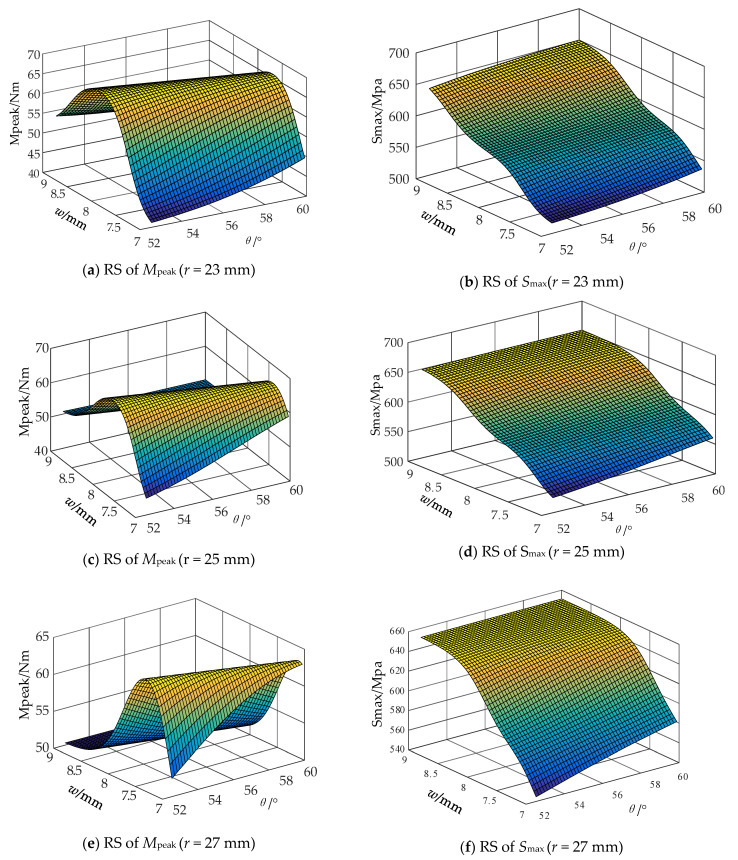
Response surfaces of the *M_peak_* and *S_max_* under different *r* value.

**Figure 13 materials-15-00306-f013:**
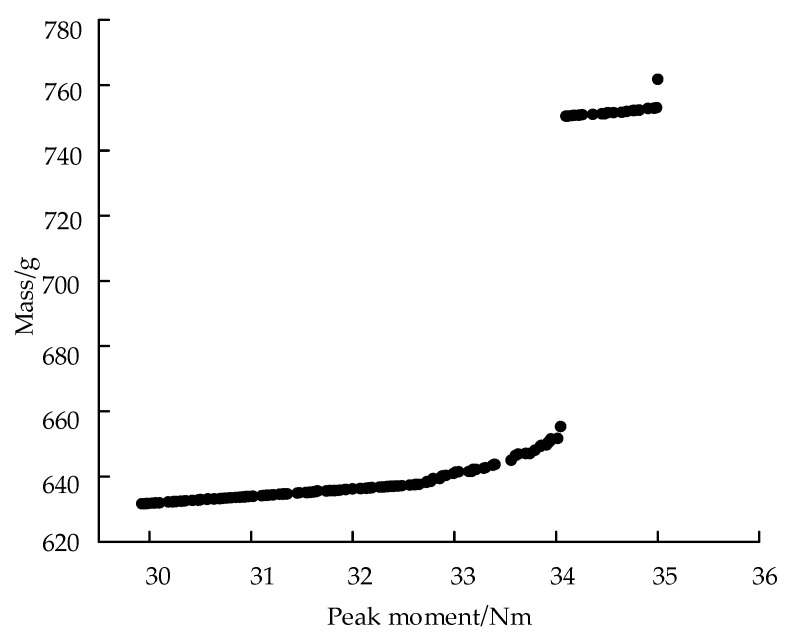
Pareto fronts of *Mass* and *M_peak_* for FLHD booms with *S_max_* ≤ 650 MPa.

**Figure 14 materials-15-00306-f014:**
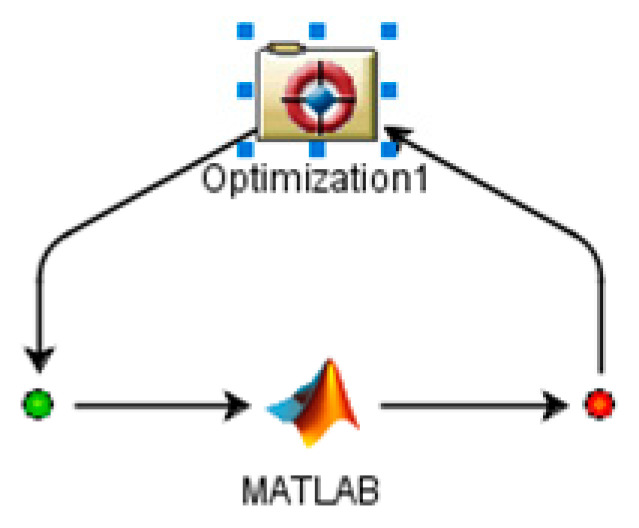
Optimization loop diagram.

**Table 1 materials-15-00306-t001:** Material properties of T800.

Material Properties	T800
Longitudinal stiffness *E*_1_/GPa	150
Transverse stiffness *E*_2_ = *E*_3_/GPa	9.4
Shear stiffness *G*_12_ = *G*_13/_GPa	9.4
In-plane shear stiffness *G*_23_/GPa	4.5
Poisson’s ratio *υ*	0.3
Density *ρ* kg/m^3^	2500

**Table 2 materials-15-00306-t002:** Sample points.

No.	*r*/mm	*θ*/°	*w*/mm	*M_peak_*/Nm	*S_max_*/MPa
1	27	52.5	7	30.01	620
2	27	52.5	8	30.02	632
3	27	52.5	9	38.32	635
4	27	55	7	27.81	632
5	27	55	8	30.88	634
6	27	55	9	33.67	654
7	27	57.5	7	26.08	702
8	27	57.5	8	31.79	713
9	27	57.5	9	32.87	675
10	27	60	7	26.44	620
11	27	60	8	38.28	679
12	27	60	9	34.43	719
13	25	52.5	7	29.86	647
14	25	52.5	8	34.94	685
15	25	52.5	9	33.89	682
16	25	55	7	25.58	663
17	25	55	8	25.62	687
18	25	55	9	29.88	675
19	25	57.5	7	27.23	674
20	25	57.5	8	27.91	671
21	25	57.5	9	28.60	674
22	25	60	7	30.03	656
23	25	60	8	28.14	649
24	25	60	9	31.60	694
25	23	52.5	7	29.45	624
26	23	52.5	8	31.18	620
27	23	52.5	9	31.35	673
28	23	55	7	37.76	704
29	23	55	8	39.33	714
30	23	55	9	28.64	730
31	23	57.5	7	30.52	681
32	23	57.5	8	32.62	695
33	23	57.5	9	29.32	729
34	23	60	7	34.57	642
35	23	60	8	35.17	736
36	23	60	9	41.01	668

**Table 3 materials-15-00306-t003:** Errors of the test sample points.

No	*r*/mm	*θ*/°	*w*/mm	*M_peak_*/Nm		RE/(%)	*S_max_*/MPa		RE/(%)
				FE Result	BPNN Result		FE Result	BPNN Result	
1	23	57	8.5	35.089	32.8801	−6.30	714	737.5714	3.30
2	24	53	7.5	30.3478	29.3259	−3.37	655	648.597	−0.98
3	24.5	56	7.3	25.534	26.013	1.88	643	677.66	5.39
4	26.5	60	7.2	24.926	25.775	3.41	657	639.709	−2.63
5	25.5	54	8.5	30.2272	29.7083	−1.72	667	691.2579	3.64

**Table 4 materials-15-00306-t004:** Optimal design of the FLHD boom.

No.	*r*/mm	*φ*/°	*w*/mm	*M_peak_*/Nm		RE/(%)	*S_max_*/MPa		RE/(%)
				FE Result	BPNN Result		FE Result	BPNN Result	
1	23.00	53.31	7.52	33.72	34.02	0.87	688	650	−5.54

## Data Availability

Not applicable.

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
