# Peer review of "Novel Four-Cell Lenticular Honeycomb Deployable Boom with Enhanced Stiffness"

_materials, 2022, doi:10.3390/ma15010306_

Round 1
Reviewer 1 Report
The paper discusses a thin-walled boom, its design, coiling dynamics, finite simulation of stress state and system optimization. Neural network based approximation of wrapping moment and stress was used. Optimized boom parameters were obtained using non-dominated sorting genetic algorithm-II.
Some possible applications of thin-walled boom for producing tribo-fatigue systems using new structural materials could be referred to in the paper:
(i) a method of experimental study of friction in a active system, (ii) state of volumetric damage of tribo-fatigue system, (iii) spatial stress-strain state of tribofatigue system in roll-shaft contact zone, (iv) modeling of the damaged state by the finite-element method on simultaneous action of contact and noncontact loads, (v) tribo-fatigue behavior of austempered ductile iron monica as new structural material for rail-wheel system, (vi) research on tensile behaviour of new structural material monica, (vii) measurement and real time analysis of local damage in wear-and-fatigue tests
Quality of formulas and figures should be enhanced.
Artefact text like “The Materials and Methods should be described with sufficient details to allow others to replicate and build on the published results” should be deleted.
What is Smax? Is it a first principal stress or von Mises stress?
It is expedient to consider errors for NN simulation not for the training samples but for the test samples that were not used in training.
The paper “Novel four-cell lenticular honeycomb deployable boom with enhanced stiffness” could be published in Materials after revision.
Reviewer 2 Report
The submitted manuscript deals with the coiling dynamic in honeycomb deployable boom. The authors study this mechanism by finite element numerical modeling and apply a neural network approach to study this problem and optimize the process.
The manuscript is poorly written and badly presented. Reading the text, the scarce care of the authors is evidenced by the presence of text from the instruction for the authors (lines 104-109, line 319), inhomogeneous and hardly readable text in the figures, missing punctuation, and numerous typos in the text.
Nevertheless, the factor which most negatively affects the present article is that the purpose and the real impact of this study are not clear. Reading this manuscript, honestly, I cannot find a clear purpose for this study.
Some more issues detected are following listed:
- The considered materials are not reported.
- The choices made (dimensions, application of the boundary conditions, and loads) in the numerical model setup are not justified. I cannot see the relation with a possible real case.
- Moreover, I cannot understand whether the analyses are steady or transient and which motions are allowed.
- In Figure 5, I can see the penetration of one of the radial rollers.
- Was the interval of 0.36 mm between boom and hub surfaces chosen arbitrarily?
- In the contours reported in Figure 7, the units are not reported.
- Which moment computed in which point does figure 8 represents?
- Methods and results are mixed in the same sections, leading to a poor explanation of both of them.
- Subsection 3.2 BPNN Surrogate Model presents basic contents in high detail, considering that this should be a scientific paper, and without mentioning references.
Reviewer 3 Report
Deployable composite structure boom seems to be one of the important techniques to develop the lightweight satellite and spacecraft. The submitted article describes the research activities to develop the lenticular honeycomb deployable boom and it is worth for publication.
However, the reviewer asks the following questions and comments, before it is finally approved for publication.
1.
In the paper, the equations and the characters in figure are not clear. The authors should revise them.
2.
The reviewer recommends the author to unify the unit in the paper. For example, thickness of each ply is described as 4.5e-5 (m) in fig. 4 and it is written as 0.045mm at p.3, line 98.
3.
The author should explain about Non-dominated sorting algorithm-II and ISIGHT in the paper.
4.
- 3, line 91, there is a description, “Two radial rollers are symmetrically positioned on both sides of the FLHD boom”. Please identify the roller name in fig. 3 for easy recognition.
5.
In fig. 3, 11 rollers seem not to be evenly positioned.
6.
p.3, line 94, there is a description, “The axial distance between reference point (RP) 13 and RP14 is 78 mm, and the radial distance between the outer arc of the circumferential roller and the edge of the roller is 25 mm”. However, RP13 is not shown in fig.3.
7.
The reviewer recommends the authors to write the dimension in fig. 3.
8.
P.4, Table 1 shows the material property of T800. What is T800? The authors should describe the name of the manufacturer of T800 and explain it.
9.
The sentence between line 104 and line 110 looks like the description in template and should be removed.
10.
P.4, line 116, there is a description, “The interaction between the rolls and flattener is set as. …..”. What is flattener?
11
P.4, line 120, there is a description, “No direct contact exists between the surface of the boom and the surface of the hub, and an interval of 0.36 mm is set to avoid stress concentration.”. Is it a technique for numerical simulation? How is the interval of 0.36mm defined?
12
The authors should describe the location of node set G and the axis in fig. 5.
13
How is the contact between tape-strings modeled?
14
The moment of RP1 is shown in fig.8. The authors should explain why the moment of RP1 is selected to explain.
15
p.5. line 151, there is a description, “While the wrapping Mpeak can lock the deployment mechanism, ….” Does Wrapping Mpeak mean that the high bending moment exist on the boom along longitudinal direction? And is it confirmed that Mpeak is located at RP1?
In addition, how much is the threshold value?
16
p.6 line 172. The author should describe why 3-15-2 system is adopted. Especially why 15 is selected in the hidden layer?
17
The authors should explain Fig.10 from the point of overfitting.
18
What is RE in fig.11?
19
p.12, line 252, there is a sentence, “The REs of Mpeak and Smax are less than 10%, ..”. The author should explain what the sentence means.
20
The reviewer recommends the authors to describe the subject in the next step and the final goal of this research in conclusion.
21
Is it the requirement of the journal to write the Author contribution?
22
Ref. 23 should be removed.
Round 2
Reviewer 1 Report
The paper is recommended for publication.
Reviewer 2 Report
The authors positively reviewed the manuscript. I am prone to propose its acceptance after these minor revisions:
- Mention in the text that carbon fibers are considered and include also the polymeric matrix used in T800.
- The penetration still appears in Figure 5. The authors should justify it also in the text, as they did in the response to the previous revision. If the penetrating rollers do not play an active role in the simulation, the authors should consider removing them from the figure.
Reviewer 3 Report
Before publication, authors should check the following points.
p.4, line 125-128
The interactions between the rollers and FLHD boom or between the hub and FLHD boom are constructed with surfaces contact as frictionless property. The interaction between the radial rollers, the hub and FLHD boom are set as a surface contact with frictionless properties.
These sentences seem to be same.
p.4, line 138-166.
Description for node set F and G are described in two times.
p.13, line 333
The word “NSGA-II” appears first in line 329. Then description the non-dominant sorting genetic algorithm (NSGA-II) should be used in line 329.
